# Selene-Ethylenelacticamides and *N*-Aryl-Propanamides as Broad-Spectrum Leishmanicidal Agents

**DOI:** 10.3390/pathogens12010136

**Published:** 2023-01-13

**Authors:** Natália Ferreira de Sousa, Helivaldo Diógenes da Silva Souza, Renata Priscila Barros de Menezes, Francinara da Silva Alves, Chonny Alexander Herrera Acevedo, Thaís Amanda de Lima Nunes, Zoe L. Sessions, Luciana Scotti, Eugene N. Muratov, Francisco Jaime Bezerra Mendonça-Junior, Klinger Antônio da Franca Rodrigues, Petrônio Filgueiras de Athayde Filho, Marcus Tullius Scotti

**Affiliations:** 1Post-Graduate Program in Natural and Synthetic Bioactive Products, Federal University of Paraíba, João Pessoa 58051-900, PB, Brazil; 2Post-Graduate Program in Chemistry, Federal University of Paraíba, João Pessoa 58051-900, PB, Brazil; 3Infectious Diseases Laboratory, Federal University of Delta of Parnaíba, Av. São Sebastião, nº 2819-Nossa Sra. de Fátima, Parnaíba 64202-020, PI, Brazil; 4Laboratory for Molecular Modeling, Division of Chemical Biology and Medicinal Chemistry, UNC Eshelman School of Pharmacy, University of North Carolina, Chapel Hill, NC 27599, USA

**Keywords:** leishmaniasis, *N*-aryl-propanamides, selene-ethylenelactamides, CADD, organic synthesis

## Abstract

The World Health Organization classifies *Leishmania* as one of the 17 “neglected diseases” that burden tropical and sub-tropical climate regions with over half a million diagnosed cases each year. Despite this, currently available anti-leishmania drugs have high toxicity and the potential to be made obsolete by parasite drug resistance. We chose to analyze organoselenides for leishmanicidal potential given the reduced toxicity inherent to selenium and the displayed biological activity of organoselenides against *Leishmania.* Thus, the biological activities of 77 selenoesters and their *N*-aryl-propanamide derivatives were predicted using robust in silico models of *Leishmania infantum, Leishmania amazonensis, Leishmania major, and Leishmania (Viannia) braziliensis*. The models identified 28 compounds with >60% probability of demonstrating leishmanicidal activity against *L. infantum*, and likewise, 26 for *L. amazonesis*, 25 for *L. braziliensis*, and 23 for *L. major*. The in silico prediction of ADMET properties suggests high rates of oral absorption and good bioavailability for these compounds. In the in silico toxicity evaluation, only seven compounds showed signs of toxicity in up to one or two parameters. The methodology was corroborated with the ensuing experimental validation, which evaluated the inhibition of the Promastigote form of the *Leishmania* species under study. The activity of the molecules was determined by the IC_50_ value (µM); IC_50_ values < 20 µM indicated better inhibition profiles. Sixteen compounds were synthesized and tested for their activity. Eight molecules presented IC_50_ values < 20 µM for at least one of the *Leishmania* species under study, with compound NC34 presenting the strongest parasite inhibition profile. Furthermore, the methodology used was effective, as many of the compounds with the highest probability of activity were confirmed by the in vitro tests performed.

## 1. Introduction

The World Health Organization classifies *Leishmania* as one of the 17 “neglected diseases” that, burden primarily tropical and sub-tropical climate regions [1]. These neglected diseases are caused by parasitic agents and are considered endemic in low-income populations. To date, they have been identified in 149 countries and are responsible for anywhere between 500,000 and 1,000,000 cases annually [2].

This infectious pathology comes from the Trypanosomatidae *Leishmania* sp protozoa [3,4] being transmitted to humans through the bite of infected female sand flies [5]. Among the existing *Leishmania* species, *Leishmania infantum* is a flagellated protozoan that causes visceral leishmaniasis in the Americas [6]. Additionally, known as kala azar, tropical splenomegaly and dundum fever, *L. infantum* induces a systemic infectious disease with long-lasting fever, enlarged liver and spleen, weight loss, weakness, reduced muscle strength, and anemias [7].

The transmission cycle of this disease initiates with the bite of the phlebotomus female straw sandfly in an infected or reservoir-type host. The mosquito aspirates parasitized macrophages or amastigotes, which are free in the blood as well as in tissues. The amastigote forms, upon reaching the insect’s midgut, transform into promastigotes. These promastigotes are flagellate and allow for rapid infectious and migratory multiplication from the intestine to the larynx of the transmitting mosquito. From the fore-gut, these forms are regurgitated to be introduced into the skin of the next host if/when the mosquito makes a new bite [8].

*L. amazonensis,* together with *L. braziliensis,* are some of the causative agents of american tegumentary leishmaniasis, which is considered one of the most important Brazilian public health endemics [9]. Its vast national distribution, severe clinical forms, difficult diagnosis, and limited treatment make this disease particularly threatening [10]. *Leishmania major* is known (according to the classification of Leishmaniasis proposed by Samuel Pessoa) as the cause of the oriental bud, and is divided into the classic (dry) type caused by *L. tropica minor*, and the wet type caused by *L. tropica major* [11].

Therapeutic agents against leishmaniasis include meglumine antimony, pentamidine isethionate and amphotericin B (deoxycholate and liposomal). Additionally, pentoxifylline was incorporated with antimony treatments in *Leishmania* mucosa cases in Brazil [12]. Mitefosine is the only oral drug available, which is available in Brazil under the Unified Health System (SUS) for the treatment of Tegumentary Leishmaniasis, according to the deliberation of Ordinance No. 56, of October 30 of 2018 [13].

Despite the few therapeutics that are currently available, new therapeutic drugs are desperately needed to address the diverse side effects, high levels of toxicity, and drug-resistance prevalent in current treatments [14,15]. In the search for alternatives, Luís and Collaborators (2019) [16] performed a virtual screening of selenoglycolicamide derivatives and validated the in silico methods experimentally through inhibition assays of promastigote *L. amazonensis*. The authors selected 12 molecules using classification models and molecular docking. Of these, seven showed experimental, biological activity against *L. amazonensis*. In another study, Plano and Collaborators (2011) [17] synthesized selenocyanate and diselenide derivatives to investigate their activity against *L. infantum*. Of the 35 compounds organically synthesized for evaluation, 13 exhibited better IC_50_ values than miltefosine and edelfosine, representing a promising new alternative for anti-leishmania activity. Figure 1 demonstrates the structures that inspired this work.

In response to the literature, this work virtually screened selenium compounds derived from ethylenelactic acid (selene-ethylenelacticamides) and evaluated their activity against the amastigote and promastigote forms of the flagellated parasites *L. amazonensis, L. infantum, L. major, and L. braziliensis.*

## 2. Materials and Methods

### 2.1. Data Collection and Curation

Chemical compounds with reported activity (pIC_50_) against *L. amazonensis* (CHEMBL612877)*, L. infantum* (CHEMBL612848)*, L. major* (CHEMBL612879), and *L.braziliensis* (CHEMBL612878) (EMBL-EBI, Wellcome Genome Campus, Cambridgeshire, England) were obtained from the CHEMBL database (https://www.ebi.ac.uk/chembl/, accessed on 10 November 2021) [18,19] to construct predictive QSAR models. The test series was composed of a total of 77 compounds of synthetic origin, obtained through organic synthesis, with 48 new compounds derived from ethylenelactic acid and called selene-ethylenelacticamides. The remaining 29 compounds corresponded to *N*-arylpropanamides (which are the intermediates in the organic synthesis) and are compounds already reported in Sherbiny and Collaborators (2016) [20] and Huang and Collaborators (2021) [21]. All compounds were selected by their chemical and biological data, adhering to the workflows elaborated by Fourches et al. [22,23]. The compounds were designed in the Marvin Sketch 18.14 software program [JChem 17.18.0.1784, 2017, ChemAxon (https://chemaxon.com/, accessed on 10 November 2021). After obtaining the design, the compounds were converted into SMILES, which constitute a personalized canonical representation, by the J Chem software (https://chemaxon.com/, accessed on 1 November 2022). The 3D structures were generated by Chemaxon Standardizer v.18.17.0, (ChemAxon, Boston, MA, USA, www.chemaxon.org, accessed on 10 November 2021). In this software, the structures were standardized through the addition of Hydrogen (H) atoms in the compounds, standardization of the aromatic ring, conversion to a 3D structure, and the removal of salts. Standardizer’s main purpose is to transform chemical structures into representations that obey certain chemical business rules to avoid inconsistencies in a chemical database. The tool is typically used in workflows where new compounds are registered, or where structure-based virtual screening is performed. There are 40 pre-defined standardizer actions available that cover among others the following issues: Adding or removing explicit H atoms; Neutralizing charged fragments or functional groups; Recognizing and converting legacy representations of functional groups (like aliases); Removing certain fragments (like water and salt counterions); 2D cleaning and expanding abbreviated groups; Unified representation of aromatic rings, tautomers and mesomers. Different from the one-click process implemented in the “Standardization of molecules”, the “Custom preprocessing” sub-module provides flexible options for users to construct customized standardization process according to their own preferences of operations and execution orders [24,25].

### 2.2. QSAR Modelling

Four prediction models were built, one each for each of the Promastigotes forms of *L. infantum, L. amazonensis, L. braziliensis* and *L. major*. Datasets were obtained from the ChEMBL Database (https://www.ebi.ac.uk/chembl/, accessed on 10 November 2021) [19]. The compounds were classified according to the pIC50 value (-log IC50 (mol/L), which represents the concentration necessary for the inhibition of 50% of the forms of the parasite under study. The databases used are described below:

The data set for *L. amazonensis,* in its Promastigote form, was composed of 235 structures, with those having a pIC50 ≥ 4.99 (118 compounds) being considered active. The compounds that were classified as inactive referred to those with pIC50 ≤ 4.97 (117 compounds), with a border of 0.3 added and removing nine compounds. The Promastigote data set consisted of 516 compounds in total, with 12 selenoglycolicamide compounds developed and experimentally validated by Luís and Collaborators (2019) [16] incorporated into this set, resulting in 528 compounds. In order to improve the classification of the elaborated model, a border was placed, in which the pIC50 value of the withdrawn compounds comprised the range of 4.78 to 4.71, totaling 32 withdrawn compounds and 496 remaining compounds. Thus, the 248 compounds classified as active presented a pIC50 ≥ 4.79, whereas the inactive compounds had pIC50 values ≤ 4.71.

The data set for *L. infantum* in its Promastigote form contained 265 compounds in total, with compounds that presented pIC50 values ≥ 4.84 (134 compounds) being active, and those with pIC50 values < 4.84 (131 compounds) being inactive.

The data set for *L. braziliensis* in its Promastigote form contained 265 compounds in total, with active pIC50 values ≥ 4.31 (135 compounds) and inactive pIC50 values < 4.30 (130 compounds).

The data set for *L. major* in its Promastigote form, with 619 compounds in total, featured 312 active compounds (pIC50 values ≥ 5.10) and 307 inactive compounds (pIC50 values < 5.08).

The Knime 3.6.2 software (Knime 3.6.2, Copyright Miner, de Konstanz Information, Zurich, Suíça, www.knime.org, accessed on 10 November 2021) was used to build and evaluate the predictive QSAR models. Due to the success of previous studies carried out in the laboratory [26,27], we chose to use 3D QSAR analysis. For this, all compounds were converted into 3D structures, saved in SDF format, and imported into Dragon 7.0 software (Kode Chemoinformatics srl, Pisa, Italy) [28] and VolSurf (Volume and Surface) [29] to obtain the descriptors. The Random Forest (RF) algorithm was used to build predictive models. The applicability domain was calculated according to the Euclidean distances present in the surveyed chemical space [27]. External cross-validation was performed to estimate the predictive power of the developed models. The performance of external models was evaluated by ROC analysis. The models were also analyzed using the MCC confusion matrix to evaluate the model globally. The complete description of the procedure performed in the Knime 3.6.2 software is shown in Figure 2.

### 2.3. VolSurf Descriptors

The Principal Component Analysis (PCA) was performed to identify the most significant variables based on the structures of the compounds. For the generation of the PCA models, the OH_2_ probe (hydrophilic probe), DRY (hydrophobic probe), N1 (amide nitrogen-hydrogen-bond donor probe) and O (carbonyl oxygen-hydrogen-bond acceptor probe) were selected [30]. The selection of the GRID force field was also performed, calculating sites of energetically favorable interactions [31]. In addition to the PCA analyses, Consensus experiments of the Principal Components Analysis (CPCA) were used to graphically evaluate the most significant blocks of descriptors, as well as the validation statistics of the variances of the first principal components (PC).

### 2.4. Homology Modelling

Sequences of enzymes and species selected in the study were obtained from the GenBank Database (National Center for Biotechnology Information, Bethesda MD, USA, https://www.ncbi.nlm.nih.gov/, accessed on 15 January 2022) [32,33], while the model structures were obtained from the Protein Data Bank library (PDB, https://www.rcsb.org/pdb/home/home.do, accessed on 15 January 2022) [34,35]. Twelve enzymes were constructed through homology models. Enzyme models were built using the molecular homology method in MODELLER 9.20 software (copyright © 2020–2020 Andrej Sali, maintained by Ben Webb in the Departments of Biopharmaceutical Sciences and Pharmaceutical Chemistry, and California Quantitative Biomedical Research Institute, Mission Bay Byers Hall, University of California, San Francisco, San Francisco, USA) [36,37,38]. Five models were generated, and the lowest energy model was chosen. The stereochemical qualities of the model were evaluated by the PSVS (Protein Structure Validation Software suite) web server (http://psvs-1_5-dev.nesg.org/, accessed on 15 January 2022), using PROCHECK [39]. PROCHECK software generates a Ramachandran graph [40], which determines which amino acid regions are allowed and not allowed in the main chain. Structural quality was assessed using VERIFY 3D software (save @ 2020 - DOE-MBI Services, http://services.mbi.ucla.edu/SAVES/, accessed on 15 January 2022) [22], and the compatibility between the protein sequence and its 3D structure, based on the chemical environment, was analyzed using WHAT IF (http://swift.cmbi.ru.nl/servers/html/index.html, accessed on 15 January 2022) [41]. Homology modeling was carried out because, with the exception of *L. infantum* and *L. major* which have two 3D structures obtained by x-ray crystallography, the other species of *Leishmania* under study (*L. amazonensis, L. major and L. braziliensis*) do not present all available mechanisms, so the homologous proteins were constructed.

The homology results can be found in Appendix A.

### 2.5. Molecular Docking Studies

Proteins were obtained from the Protein Data Bank (PDB) library (https://www.rcsb.org/, accessed on 1 November 2022) [35]. The targets were selected through bibliographical research on the mechanism of action involved in the inhibition of the parasites under study, as well as considering their structural similarity. Redocking was performed prior to validate the docking simulation. Both procedures were performed using Molegro Virtual Docker (MVD) v.6.0.1 software [42]. Enzymes and compounds were prepared according to predefined parameters in the software. In the coupling procedure (linker-enzyme), a Grid of 15 Å radius and 0.30 resolution was used, which involved the site of the binding site, defined by a known ligand for each enzyme. A model was generated to perform and evaluate the fit with expected characteristics between the ligand and the enzyme, using the MOLDOCK Score (GRID) algorithm with the scoring function and search algorithm, corresponding to Moldock. The MolDock scoring function further improves these scoring functions with a new hydrogen bonding term and new charge schemes. The docking scoring function, Escore, is defined by the following energy terms:E_score_ = E_inter_ + E_intra_

The visualization of the established interactions was performed in the Discovery Studio Visualizer program, Biovia, 2020 (https://www.3dsbiovia.com/, accessed on 22 April 2022) [43].

Molecular Docking results can be found in Appendix A.

### 2.6. Predicting In Silico ADMET Properties

Calculation of ADMET parameters and toxicity risk predictions were performed using the OSIRIS Property Explorer software (Idorsia Pharmaceuticals Ltd., Allschwil, Switzerland, https://www.organic-chemistry.org/prog/peo/, accessed on 15 January 2022). The compounds were evaluated for their risk of presenting toxicity in three parameters: mutagenicity, skin irritability and reproductive toxicity. The risk assessment is carried out qualitatively so that the compounds are classified as either low risk or high risk [44,45]. For the absorption study, factors such as membrane permeability and intestinal absorption were included. We also limited our investigation to compounds that did not exceed more than two violations of Lipinski’s rule and for which the logP values were in consensus.

### 2.7. In Vitro Assay

*L. infantum, L. amazonensis, L. major* and *L. braziliensis* were maintained in vitro as promastigotes at 26 ◦C in supplemented Schneiderinsectmedium (20% SFB, 100 U/mL penicillin and 100 µg/mL streptomycin, pH 7), as described by Rodrigues et al. [46]. The growth inhibition assay for the parasites was performed using promastigote forms in the logarithmic phase, which were grown in 96-well plates containing supplemented Schneider insect medium and 1 × 10^6^ parasites/mL. This was performed in triplicate using different concentrations of lignans (1.56–12.5 µM), reference meglumine antimoniate drugs (200–40,000 µM), and amphotericin B (0.031–2 µM). The negative control contained neither the reference nor the tested compounds. The culture plates were maintained in a biological oxygen demand incubator (Eletrolab EL202, São Paulo, Brazil), at 26 °C, for axenic promastigotes. After 2 days under these conditions, 10 µL of 3-(4,5-Dimethylthiazol-2-yl)-2,5-Diphenyltetrazolium Bromide (MTT, 5 mg/mL) was added to each well, and the cell culture plates were incubated for 4 h before adding 50 µL of 10% sodium dodecyl sulfate (SDS) solution, to solubilize the formazan crystals. The optical density of the culture was measured in a microplate spectrophotometer reader at 540 nm (Biosystems ELx800 model, Curitiba, Brazil).

### 2.8. Molecular Dynamics

MD simulations were performed to estimate the flexibility of interactions between proteins and ligands using GROMACS 5.0 software (European Union Horizon 2020 Programme, Stockholm, Sweden) [47,48]. The protein and ligand topologies were also prepared using the GROMOS96 54a7 force field. The MD simulation was performed using the SPC water model of point load, extended in a cubic box [49]. The system was neutralized by the addition of ions (Cl^−^ and Na^+^) and minimized to remove bad contacts between complex molecules and the solvent. The system was also balanced at 300K, using the 100 ps V-rescale algorithm, represented by NVT (constant number of particles, volume, and temperature), followed by equilibrium at 1 atm of pressure using the Parrinello-Rahman algorithm as the NPT (constant pressure particles and temperature), up to 100 ps. DM simulations were performed in 5,000,000 steps, each at 10 ns. To determine the flexibility of the structure and whether the complex is stable close to the experimental structure, RMSD values of all Cα atoms were calculated relative to the starting structures. RMSF values were also calculated to understand the roles of nearby residues in the receptor binding site. The RMSD and RMSF graphs were generated in Grace software (Grace Development Team, http://plasma-gate.weizmann.ac.il/Grace/, accessed on 11 November 2022) and the protein and ligands were visualized in the UCSF Chimera [50]. The compound submitted to this simulation was NC34, since this molecule presented IC50 values < 10 µM in the four species of Leishmania under study, as well as the correspondence between the probability value obtained in the virtual screening and the result of the biological tests in *L. infantum* and *L. amazonensis*. Further, it displayed affinity values close to the reference ligand for CYP-51 enzymes (PDB: 3L4D) of *L. infantum* (−112.01 Kcal/mol and DSC of −0.30), and for the Dihydroorotate Dehydrogenase enzyme of *L. amazonensis* (−135.69 Kcal/mol and DSC of −2.68).

### 2.9. Major Signs Characterizing N-Arylpropanamide Compounds

The 16 compounds below were chosen for synthesis due to the ease of obtaining them, as well as a series of selenium compounds called selenoglycolicamides, which have similar structures have already been experimentally promising in Leishmania studies by Luis and Collaborators (2019) [16]. They were also chosen based on in silico ADME predictions in which the Selene-ethylenelacticamide derivatives showed high percentages of oral absorption. None of the compounds selected for synthesis violated Lipinski’s rule, confering sufficient bioavailability.

3-chloro-*N*-phenylpropanamide (NC01): Yield: 80%.P.F: 119–121 °C. IV (ATR, cm^−1^): 3300 (N-H); 3144, 3093 (=C-H, Ar); 2983, 2920 (C-H, aliphatic); 1658 (-N-CO); 1606 e 1442 (C=C, Ar). ^1^H NMR (500 MHz, CDCl_3_) ***δ*** 7.76 (s, 1H, N-H), 7.51 (d, *J* = 7.8 Hz, 2H, H-5 e H-5′), 7.31 (t, *J* = 7.9 Hz, 2H, H-6 e H-6′), 7.12 (t, *J* = 7.4 Hz, 1H, H-7), 3.85 (t, *J* = 6.4 Hz, 2H, H-3), 2.80 (t, *J* = 6.4 Hz, 2H, H-2). ^13^C NMR (126 MHz, CDCl_3_) ***δ*** 168.13 (C-1), 137.57 (C-4), 129.13 (C-6 e C-6′), 124.83 (C-7), 120.40 (C-5 e C-5′), 40.51 (C-3), 40.03 (C-2).

3-chloro-*N*-(4-methylphenyl)propanamide (NC02): Yield: 85%. P.f.; 121–123 °C. IV (ATR, cm^−^^1^): 3238 (N-H); 3120, 3072 (=C-H, Ar); 2983, 2916, 2862 (C-H, aliphatic); 1643 (-N-CO); 1602 and 1510 (C=C, Ar).^1^H NMR (500 MHz, CDCl_3_) ***δ*** 7.39 (d, *J* = 8.3 Hz, 2H, H-5 e H-5′), 7.32 (s, 1H, N-H), 7.12 (d, *J* = 8.2 Hz, 2H, H-6 e H-6′), 3.88 (t, *J* = 6.5 Hz, 2H, H-3), 2.79 (t, *J* = 6.5 Hz, 2H, H-2), 2.31 (s, 3H, CH_3_). ^13^C NMR (126 MHz, CDCl_3_) ***δ*** 167.70 (C-1), 134.99 (C-7), 134.53 (C-4), 129.67 (C-6 e C-6′), 120.33 (C-5 e C-5′), 40.67 (C-3), 40.09 (C-2), 21.01 (CH_3_).

3-chloro-*N*-(4-methoxyphenyl)propanamide (NC07): Yield: 75%. P.f.; 124–126 °C. IV (ATR, cm^−1^): 3278 (N-H); 3138, 3005 (=C-H, Ar); 2953, 2904 (C-H, Aliphatic); 1651 (-N-CO); 1602 and 1508 (C=C, Ar), 1240 e 1029 (C-O). ^1^H NMR (400 MHz, CDCl_3_) ***δ*** 7.61 (s, 1H, N-H), 7.39 (d, *J* = 9.0 Hz, 2H, H-5 e H-5′), 6.83 (d, *J* = 9.0 Hz, 2H, H-6 e H-6′), 3.85 (t, *J* = 6.5 Hz, 2H, H-3), 3.77 (s, 3H, OCH_3_), 2.77 (t, *J* = 6.5 Hz, 2H, H-2). ^13^C NMR (101 MHz, CDCl_3_) ***δ*** 167.93 (C-1), 156.84 (C-7), 130.66 (C-4), 122.33 (C-5), 114.29 (C-6 e C-6′), 55.60 (OCH_3_), 40.37 (C-3), 40.16 (C-2).

3-chloro-*N*-(4-chlorophenyl)propanamide (NC08): Yield: 70%, P.f.; 123–125 °C. IV (ATR, cm^−1^): 3290 (N-H); 3126, 3072 (=C-H, Ar); 2966 (C-H, aliphatic); 1653 (-N-CO); 1606 and 1489 (C=C, Ar).^1^H NMR (400 MHz, CDCl_3_) ***δ*** 7.78 (s, 1H, N-H), 7.45 (d, *J* = 8.8 Hz, 2H, H-5 e H-5′), 7.26 (d, *J* = 8.8 Hz, 2H, H-6 e H-6′), 3.85 (t, *J* = 6.4 Hz, 2H, H-3), 2.80 (t, *J* = 6.4 Hz, 2H, H-2). ^13^C NMR (101 MHz, CDCl_3_) ***δ*** 168.18 (C-1), 136.12 (C-4), 129.16 (C-6 e C-6′), 128.43 (C-7), 121.61 (C-5 e C-5′), 40.45 (C-3), 39.91 (C-2).

3-chloro-*N*-(4-isopropylphenyl)propanamide (NC09): Yield: 77%. P.f.; 124–126 °C. IV (ATR, cm^−1^): 3296 (N-H); 3130, 3076 (=C-H, Ar); 2958, 2866 (C-H, aliphatic); 1654 (-N-CO); 1604 and 1413 (C=C, Ar). ^1^H NMR (400 MHz, CDCl_3_) ***δ*** 7.63 (s, 1H, N-H), 7.43 (d, *J* = 8.5 Hz, 2H, H-5 e H-5′), 7.17 (d, *J* = 8.4 Hz, 2H, H-6 e H-6′), 3.86 (t, *J* = 6.5 Hz, 2H, H-3), 2.87 (m, 1H, CH), 2.79 (t, *J* = 6.5 Hz, 2H, H-2), 1.22 (d, *J* = 6.4 Hz, 6H, CH_3_). ^13^C NMR (101 MHz, CDCl_3_) ***δ*** 167.94 (C-1), 145.57 (C-7), 135.28 (C-4), 127.02 (C-5 e C-5′), 120.51 (C-6 e C-6′), 40.53 (C-3), 40.13 (C-2), 33.73 (CH), 24.10 (CH_3_).

3-chloro-*N*-(4-nitrophenyl)propanamide (NC13): Yield: 74%.P.f.; 124–126 °C. IV (ATR, cm^−1^): 3350 (N-H); 3107 (=C-H, Ar); 2912 (C-H, aliphatic); 1703 (-N-CO); 1593 (C=C, Ar), 1500 e 1319 (=C-NO_2_^1^H NMR (400 MHz, DMSO-*d*_6_) ***δ*** 10.68 (s, 1H, N-H), 8.22 (d, *J* = 9.2 Hz, 2H, H-6 e H-6′), 7.84 (d, *J* = 9.3 Hz, 2H, H-5 e H-5′), 3.89 (t, *J* = 6.2 Hz, 2H, H-3), 2.90 (t, *J* = 6.2 Hz, 2H, H-2). ^13^C NMR (101 MHz, DMSO-*d*_6_) ***δ*** 169.12 (C-1), 145.05 (C-7), 142.31 (C-4), 125.06 (C-6 e C-6′), 118.84 (C-5 e C-5′), 40.45 (C-3), 39.43 (C-2).

Ethyl 4-(3-chloropropanamide) benzoate: Yield (NC18): 71%. P.f.; 124–126 °C. IV (ATR, cm^−1^): 3367 (N-H); 3111, 3039 (=C-H, Ar); 2987 (C-H, aliphatic); 1685 (-N-CO e O-CO); 1597 e 1406 (C=C, Ar), 1271 e 1107 (C-O). ^1^H NMR (400 MHz, CDCl_3_) ***δ*** 8.11 (s, 1H, N-H), 7.98 (d, *J* = 8.6 Hz, 2H, H-6 e H-6′), 7.62 (d, *J* = 8.5 Hz, 2H, H-5 e H-5′), 4.35 (q, *J* = 7.1 Hz, 2H, CH_2_), 3.86 (t, *J* = 6.4 Hz, 2H, H-3), 2.85 (t, *J* = 6.4 Hz, 2H, H-2), 1.38 (t, *J* = 7.1 Hz, 3H, CH_3_). ^13^C NMR (101 MHz, CDCl_3_) ***δ*** 168.31 (C-1), 166.29 (C=O), 141.76 (C-7), 130.78 (C-4), 126.21 (C-6 e C-6′), 119.15 (C-5 e C-5′), 61.03 (CH_2_), 40.45 (C-3), 39.64 (C-2), 14.32 (CH_3_).

3-chloro-*N*-(4-bromophenyl)propanamide (NC19): Yield: 72%. P.f.; 124–126 °C. IV (ATR, cm^−1^): 3290 (N-H); 3138, 3070 (=C-H, Ar); 2964 (C-H, aliphatic); 1656 (-N-CO); 1604 e 1487 (C=C, Ar). ^1^H NMR (400 MHz, CDCl_3_) ***δ*** 7.70 (s, 1H, N-H), 7.41 (s, 4H, H-5, H-5′, H-6 e H-6′), 3.85 (t, *J* = 6.4 Hz, 2H, H-3), 2.80 (t, *J* = 6.4 Hz, 2H, H-2). C NMR (101 MHz, CDCl_3_) ***δ*** 168.12 (C-1), 136.62 (C-4), 132.13 (C-6 e C-6′), 121.87 (C-5 e C-5′), 117.49 (C-7), 40.51 (C-3), 39.89 (C-2).

### 2.10. Main Identification Signs of Selene-Ethylenelacticamides and Reaction Yield

*N*-phenylbenzoselene-ethylenelactamide (NC30): Yield: 60%. P.F: 139–141 °C. IV (ATR, cm^−1^): 3296 (N-H); 3061, 3032 (=C-H, Ar); 2949 (C-H, aliphatic); 1670 (-Se-CO); 1651 (-N-CO); 1595 e 1444 (C=C, Ar). ^1^H NMR (400 MHz, CDCl_3_) δ 7.88 (dd, *J* = 8.3, 1.2 Hz, 2H, H-3 e H-3′), 7.65–7.56 (m, 2H, H-5 e N-H), 7.53 (d, *J* = 7.9 Hz, 2H, H-10 e H-10′), 7.44 (t, *J* = 7.7 Hz, 2H, H-4 e H-4′), 7.33–7.28 (m, 2H, H-11 e H-11′), 7.09 (t, *J* = 7.4 Hz, 1H, H-12), 3.38 (t, *J* = 6.9 Hz, 2H, H-6), 2.88 (t, *J* = 6.9 Hz, 2H, H-7). ^13^C NMR (101 MHz, CDCl_3_) δ 195.63 (C-1), 169.80 (C-8), 138.90 (C-9), 137.86 (C-2), 133.98 (C-5), 129.10 (C-3 e C-3′), 128.98 (C-4 e C-4′), 127.29 (C-11 e C-11′), 124.49 (C-12), 120.06 (C-10 e C-10′), 38.50 (C-7), 20.30 (C-6).

*N-*(4-methylphenyl)benzoselene-ethylenelactamide (NC31): Yield: 59%.P.F: 128–130 °C. IV (ATR, cm^−1^): 3296 (N-H); 3064 (=C-H, Ar); 2954, 2887 (C-H, aliphatic); 1674 (-Se-CO); 1658 (-N-CO); 1600 e 1446 (C=C, Ar). ^1^H NMR (500 MHz, CDCl_3_) δ 7.89 (d, *J* = 7.3 Hz, 2H, H-3 e H-3′), 7.59 (t, *J* = 7.4 Hz, 1H, H-5), 7.48–7.43 (m, 3H, H-4, H-4′ e N-H), 7.40 (d, *J* = 8.3 Hz, 2H, H-10 e H-10′), 7.11 (d, *J* = 8.2 Hz, 2H, H-11 e H-11′), 3.38 (t, *J* = 6.9 Hz, 2H, H-6), 2.87 (t, *J* = 6.9 Hz, 2H, H-7), 2.30 (s, 3H, CH_3_). ^13^C NMR (126 MHz, CDCl_3_) δ 195.58 (C-1), 169.61 (C-8), 138.96 (C-12), 135.30 (C-2), 134.13 (C-9), 133.95 (C-5), 129.59 (C-3 e C-3′), 128.98 (C-4 e C-4′), 127.30 (C-11 e C-11′), 120.15 (C-10 e C-10′), 38.49 (C-7), 20.98 (CH_3_), 20.40 (C-6).

*N-*(4-methoxyphenyl)benzoselene-ethylenelactamide (NC34): Yield: 55%. P.F: 151–153 °C. IV (ATR, cm^−1^): 3300 (N-H); 3022 (=C-H, Ar); 2962 (C-H, aliphatic); 1670 (-Se-CO); 1651 (-N-CO); 1523 e 1408 (C=C, Ar), 1247 e 1029 (C-O). ^1^H NMR (500 MHz, DMSO-*d*_6_) ***δ*** 9.84 (s, 1H, N-H), 7.88 (dd, *J* = 8.4, 1.2 Hz, 2H, H-3 e H-3′), 7.75–7.66 (m, 1H, H-5), 7.56 (t, *J* = 7.8 Hz, 2H, H-4 e H-4′), 7.49 (d, *J* = 9.1 Hz, 2H, H-10 e H-10′), 6.86 (d, *J* = 9.1 Hz, 2H, H-11 e H-11′), 3.71 (s, 3H, OCH_3_), 3.28 (t, *J* = 6.9 Hz, 2H, H-6), 2.82 (t, *J* = 6.9 Hz, 2H, H-7). ^13^C NMR (126 MHz, DMSO-*d*_6_) ***δ*** 194.95 (C-1), 169.56 (C-8), 155.60 (C-12), 138.79 (C-2), 134.66 (C-5), 132.64 (C-9), 129.74 (C-3 e C-3′), 127.18 (C-4 e C-4′), 121.08 (C-10 e C-10′), 114.27 (C-11 e C-11′), 55.59 (OCH_3_), 37.03 (C-7), 20.81 (C-6).

*N-*(4-isopropylphenyl)benzoselene-ethylenelactamide (NC40): Yield: 58%. P.F: 124–126 °C. IV (ATR, cm^−1^): 3294 (N-H); 3037 (=C-H, Ar); 2958, 2926 (C-H, aliphatic); 1668 (-Se-CO); 1654 (-N-CO); 1593 e 1409 (C=C, Ar). ^1^H NMR (400 MHz, CDCl_3_) δ 7.89 (dd, *J* = 8.3, 1.1 Hz, 2H, H-3 e H-3′), 7.62–7.55 (m, 2H, H-5 e N-H), 7.47–7.42 (m, 4H, H-4, H-4′, H-10 e H-10′), 7.16 (d, *J* = 8.5 Hz, 2H, H-11 e H-11′), 3.38 (t, *J* = 6.9 Hz, 2H, H-6), 2.91 – 2.82 (m, 3H, H-7 e CH), 1.22 (d, *J* = 6.9 Hz, 6H, CH_3_). ^13^C NMR (101 MHz, CDCl_3_) δ 195.61 (C-1), 169.67 (C-8), 145.22 (C-12), 138.93 (C-2), 135.52 (C-9), 133.95 (C-5), 128.97 (C-3 e C-3′), 127.29 (C-4 e C-4′), 126.97 (C-11 e C-11′), 120.24 (C-10 e C-10′), 38.42 (C-7), 33.70 (CH), 24.12 (CH_3_), 20.42 (C-6).

*N-*(4-chlorophenyl)benzoselene-ethylenelactamide (NC36): Yield: 67%. P.F: 144–146 °C. IV (ATR, cm^−1^): 3360 (N-H); 3111, 3055 (=C-H, Ar); 2958 (C-H, aliphatic); 1689 (-Se-CO); 1658 (-N-CO); 1593 e 1489 (C=C, Ar); 1087 (=C-Cl). ^1^H NMR (400 MHz, DMSO-*d*_6_) ***δ*** 10.14 (s, 1H, N-H), 7.87 (dd, *J* = 8.3, 1.2 Hz, 2H, H-3 e H-3′), 7.70 (t, *J* = 6.8 Hz, 1H, H-5), 7.61 (d, *J* = 8.9 Hz, 2H, H-10 e H-10′), 7.55 (t, *J* = 7.8 Hz, 2H, H-4 e H-4′), 7.34 (d, *J* = 8.9 Hz, 2H, H-11 e H-11′), 3.27 (t, *J* = 6.8 Hz, 2H, H-6), 2.87 (t, *J* = 6.8 Hz, 2H, H-7). ^13^C NMR (101 MHz, DMSO-*d*_6_) ***δ*** 194.50 (C-1), 169.91 (C-8), 138.33 (C-2), 137.98 (C-9), 134.30 (C-5), 129.36 (C-4 e C-4′), 128.66 (C-11 e C-11′), 126.80 (C-3 e C-3′), 126.77 (C-12), 120.64 (C-10 e C-10′), 36.78 (C-7), 20.13 (C-6).

*N-*(4-bromophenyl)benzoselene-ethylenelactamide (NC53): Yield: 54%. P.F: 150–152 °C. IV (ATR, cm^−1^): 3358 (N-H); 3113, 3055 (=C-H, Ar); 2958, 2918 (C-H, aliphatic); 1689 (-Se-CO); 1656 (-N-CO); 1595 e 1487 (C=C, Ar); 1070 (=C-Br). ^1^H NMR (500 MHz, DMSO-*d*_6_) ***δ*** 10.12 (s, 1H, N-H), 7.87 (dd, *J* = 8.3, 1.1 Hz, 2H, H-3 e J-3′), 7.70 (t, *J* = 7.4 Hz, 1H, H-5), 7.55 (m, 4H, H-4, H-4′, H-10 e H-10′), 7.47 (d, *J* = 8.9 Hz, 2H, H-11 e H-11′), 3.27 (t, *J* = 6.8 Hz, 2H, H-6), 2.87 (t, *J* = 6.8 Hz, 2H, H-7). ^13^C NMR (126 MHz, DMSO-*d*_6_) ***δ*** 194.42 (C-1), 169.87 (C-8), 138.35 (C-2), 138.29 (C-9), 134.23 (C-5), 131.51 (C-11 e C-11′), 129.25 (C-3 e C-3′), 126.74 (C-4 e C-4′), 121.00 (C-10 e C-10′), 114.73 (C-12), 36.78 (C-7), 20.09 (C-6).

*N-*(4-nitrophenyl)benzoselene-ethylenelactamide (NC41): Yield: 59%. P.F: 180–182 °C. IV (ATR, cm^−1^): 3348 (N-H); 3116, 3076 (=C-H, Ar); 2999 (C-H, aliphatic); 1697 (-N-CO); 1643 (-Se-CO); 1593 e 1402 (C=C, Ar); 1504 e 1334 (=C-NO_2_). ^1^H NMR (500 MHz, DMSO-*d*_6_) ***δ*** 10.61 (s, 1H, N-H), 8.20 (d, *J* = 9.3 Hz, 2H, H-11 e H-11′), 7.87 (dd, *J* = 8.4, 1.2 Hz, 2H, H-3 e H-3′), 7.82 (d, *J* = 9.3 Hz, 2H, H-10 e H-10′), 7.70 (t, *J* = 7.4 Hz, 1H, H-5), 7.55 (t, *J* = 7.8 Hz, 2H, H-4 e H-4′), 3.29 (t, *J* = 6.8 Hz, 2H, H-6), 2.95 (t, *J* = 6.8 Hz, 2H, H-7). ^13^C NMR (126 MHz, DMSO-*d*_6_) ***δ*** 194.36 (C-1), 170.79 (C-8), 145.10 (C-9), 142.17 (C-12), 138.26 (C-2), 134.28 (C-5), 129.32 (C-3 e C-3′), 126.77 (C-4 e C-4′), 124.99 (C-11 e C-11′), 118.74 (C-10 e C-10′), 37.01 (C-7), 19.80 (C-6).

Ethyl (4-benzoselene-ethylenelacticamide) benzoate (NC51): *Yield: 56%.* P.F: 161–163 °C. IV (ATR, cm^−1^): 3334 (N-H); 3049 (=C-H, Ar); 2976, 2916 (C-H, aliphatic); 1685 (-N-CO e O-CO); 1664 (-Se-CO); 1591 and 1406 (C=C, Ar); 1274 e 1109 (C-O). ^1^H NMR (400 MHz, CDCl_3_) ***δ*** 7.98 (d, *J* = 8.8 Hz, 2H, H-3 e H-3′), 7.91 (s, 1H, N-H), 7.87 (dd, *J* = 8.4, 1.2 Hz, 2H, H-11 e H-11′), 7.64–7.57 (m, 3H, H-5, H-10 e H-10′), 7.44 (t, *J* = 7.8 Hz, 2H, H-4 e H-4′), 4.35 (q, *J* = 7.1 Hz, 2H, OCH_2_), 3.38 (t, *J* = 6.9 Hz, 2H, H-6), 2.92 (t, *J* = 6.9 Hz, 2H, H-7), 1.37 (t, *J* = 7.1 Hz, 3H, CH_3_). ^13^C NMR (101 MHz, CDCl_3_) ***δ*** 195.52 (C-1), 169.96 (C-8), 166.20 (C=O), 141.92 (C-9), 138.68 (C-2), 133.98 (C-5), 130.78 (C-11 e C-11′), 128.92 (C-3 e C-3′), 127.20 (C-4 e C-4′), 125.98 (C-12), 118.86 (C-10 e C-10′), 60.92 (OCH_2_), 38.57 (C-7), 19.92 (C-6), 14.35 (CH_3_).

## 3. Results and Discussion

### 3.1. Compounds in Study

The 77 molecules in this study were obtained through synthesis, with 48 new compounds, referred to as selene-ethylenelacticamides, derived from ethylenelactic acid (30–77). The remaining 29 compounds, *N*-arylpropanamides derivatives, are already reported by Sherbiny and Collaborators (2016) [20] and Huang (2021) [21] (1–29). Figure 3 displays the structure of each compound.

### 3.2. Quantitative Structure-Activity Relationship (QSAR) Modeling

Eight classification models were developed to perform the ligand-based virtual screening. The models used Random Forest (FR) algorithms as well as Dragon 7 [28] and VolSurf [29,30] to calculate the chemical descriptors used to analyze the activities of the compounds in *L. amazonensis, L. infantum, L. braziliensis* and *L. major*.

The RF models were evaluated for their predictive capabilities using specificity, sensitivity, positive predictive value (PPV) and negative predictive value (NPV). The performance and robustness of the models were evaluated using the Receiver Operating Characteristic Curve (ROC) and the Mathews Correlation Coefficient (MCC). Table 1 details the parameters of the four models developed with the Dragon 7 descriptors and Table 2 details the parameters found with the VolSurf descriptors.

For both of the descriptors, the cross-validation confirmed the excellent model performance, with a precision higher than 71% in all models developed. The ROC values for the depicted models were no lower than 0.8, and in consideration with the other statistics, the models were considered robust and predictive.

Given these results, the bank of selenium compounds and *N*-aryl-propanamide intermediates were then screened to select compounds that are predicted to be active against *L. amazonensis, L. infantum, L. major and L. braziliensis*.

#### Consensus Analysis

Consensus analysis was run on the generated models to increase the reliability of the independent model predictions [51,52]. After obtaining the descriptors and statistics, a consensus analysis was performed between the elaborated models. The calculation included the use of the following equation:$$p=\frac{\left(VNa\times Pa\right)+\left(VNb\times Pb\right)}{VNa+VNb}$$
where:

*VNa* = Model’s true negative rate a;

*VNb* = Model’s true negative rate b;

*Pa* = Probability of model positive compounds a;

*Pb* = Probability of model positive compounds b.


*Leishmania amazonensis*


The model built on VolSurf descriptors for *L. amazonensis* was able to predict all but two compounds within the applicability domain (NC63 and NC75). Furthermore, the model classified 39 molecules with a probability of activity above 50%. Selene-ethylenelactamide derivatives NC36 (*p* = 84%), NC34 (*p* = 81%), NC53 (*p* = 81%), NC58 (*p* = 77%) and NC30 (*p* = 77%) were selected for leishmanicidal behavior, as well as NC31(*p* = 77%), NC40 (*p* = 76%), NC51 (*p* = 66%), and NC41 (*p* = 61%).

The model using Dragon 7.0 descriptors classified 33 molecules with a probability of activity above 50% and featured two molecules above 70%. Outside of these, 17 compounds fell outside the AD. The compounds with the highest activity prediction were NC67 (*p* = 73%), NC58 (*p* = 72%), NC38 (*p* = 68%), NC37 (*p* = 68%) and NC42 (*p* = 63%). NC34, NC40, NC41, NC53, NC36, and NC30 were all above 53% and were also selected for testing due to their ease of availability.

It was observed that both descriptors classified a few molecules similarly, and as these are analogous compounds, a pattern was elucidated. Namely, NC58 was predicted in both models above 70%, NC36 by VolSurf at 84%, and NC67 in Dragon 7 at 73%. This suggests the strong influence of the bromine atom (Br) and halogens in the structures.

The consensus of these models resulted in 38 molecules with activity above 50%. The strongest candidates within the tested applicability domain were NC58 (*p* = 74%), NC34 (*p* = 72%), NC76 (*p* = 71%), NC36 (*p* = 70%) e NC53 (*p* = 70%). Additionally, NC40, NC30, NC51, NC41, and NC31 showed > 61% probability upon consensus analysis. It is worth noting that molecules with electron-withdrawing groups (EWGs) in the benzene ring closest to the selenium atom (such as molecule NC58) have greater probabilities of activity so long as it did not also have an EWG in the farthest ring. We hypothesize this is due to the lower contribution of aromatic amides in the presence of EWGs.


*Leishmania infantum*


The model utilizing VolSurf descriptors selected 57 molecules with a probability of activity > 50%, one of which displayed an activity probability above 80%. NC38, NC50, NC63 and NC75 were not within the applicability domain. The most likely molecules were NC66 (*p* = 81%), NC77 (*p* = 77%), NC76 (*p* = 77%), NC58 (*p* = 76%) and NC69 (*p* = 76%). Additionally, NC40 (*p* = 75%), NC51 (*p* = 71%), NC53 (*p* = 71%), NC36 (*p* = 65%), NC08 (*p* = 62%), NC19 (*p* = 61%), as well as NC02, NC01, NC09, NC31, NC07, NC34, and NC30 (55% < *p* < 59%), were included in the following investigation, due to the ease of obtaining these substances.

The Dragon 7.0 based models classified 42 molecules as potentially active, with two molecules registering between 80 and 84%. Twenty-two compounds were outside the applicability domain. The most likely compounds comprised, respectively, NC01 (*p* = 82%), NC08 (*p* = 79%), NC06 (73%), NC05 (*p* = 72%) and NC03 (71%). In addition to these compounds, high probabilities were recorded for NC07 (59%), NC13 (58%), NC53 (55%), and NC41 (50%).

The two descriptors showed a different classification pattern in this species of Leishmania, and it was observed that for the VolSurf descriptors the molecule with the highest probability was NC66 (*p* = 81%), a selenium compound with bulky (almost globular), more electronegative groups in its structure. The Dragon 7 descriptors, on the other hand, showed greater activity in the compounds of the *N*-aryl propanamides series, where the compound with the highest probability was NC01 (*p* = 82%).

In the consensus model, 55 molecules exceeded 50% probability. The strongest candidates are NC40 (*p* = 74%), NC53 (*p* = 71%), NC66 (*p* = 69%), NC31 (*p* = 68%) e NC51 (*p* = 68%). Other compounds of interest that were selected to move forward include NC36 (*p* = 68%), NC34 (*p* = 67%), NC19 (*p* = 60%), NCO2 (*p* = 59%), NC41 (*p* = 59%), and NC13 (*p* = 52%). Of the molecules that showed greater probability, all had a group that offered an inductive effect, except NC53, which presented a bromine atom in the ring farthest from the selenium. In this compound, the inductive effect is likely lower due to the high atomic radius of the bromine atom. Electronegative groups in the para position also reduced these inductive effects, and therefore decreased activity (NC51). Except for NC66 (whose inductive effect is decreased by two ortho electron donating groups), it also appears that activity probability increases when the benzene ring closest to the selenium was not substituted.


*Leishmania braziliensis*


For the promastigote form of *L. braziliensis*, the VolSurf-based model identified 24 molecules with a probability of activity above 50%. Among these, three had activity probabilities between 71 and 74%, and only two molecules were outside the applicability domain (NC38 and NC75). The molecules with the strongest prediction were selene-ethylenelactamide derivatives NC65 (*p* = 77%), NC28 (*p* = 75%), NC76 (*p* = 71%), NC74 (*p* = 69%) and NC70 (*p* = 68%), as well as NC51 (*p* = 58%).

The Dragon 7.0 model selected 40 molecules with a probability of activity above 50%. The five highest scoring compounds showed probability above 70% (NC65 (*p* = 79%), NC66 (*p* = 76%), NC49 (*p* = 75%), NC28 (*p* = 72%), NC54 (*p* = 70%)), while 27 compounds were not within the AD. Other compounds selected for testing include NC35 (*p* = 67%), NC34 (*p* = 67%), NC40 (*p* = 67%), NC36 (*p* = 66%), NC41 (*p* = 64%), NC51 (*p* = 64%), NC53 (*p* = 63%), NC31(*p* = 63%), and NC30 (*p* = 62%).

For this species, it was observed that the model elaborated with VolSurf descriptors was more restrictive, since it classified only 24 molecules, while the model elaborated with Dragon 7.0 descriptors classified almost twice as many compounds. Dragon 7.0 software offers a far superior number of descriptors compared to VolSurf descriptors. Despite the difference in the number of classified compounds, the importance of the methoxy groups as substituents was noticeable, since NC65 was classified as the compound with the highest probability in the two types of evaluated descriptors, corresponding to *p* = 77% in the VolSurf descriptors and *p* = 79 % in Dragon7 descriptors.

From the consensus model, 25 molecules exceeded 50% activity probability. The molecules with the highest probability were: NC65 (*p* = 78%), NC28 (*p* = 73%), NC66 9 (*p* = 72%), NC54 (*p* = 65%) and NC49 (*p* = 63%). Compounds NC51, NC40, NC53, NC41, and NC13 delivered activity probabilities between 52% and 61% and were therefore also selected for testing.


*Leishmania major*


The model developed using VolSurf descriptors resulted in 24 molecules with an activity probability > 50% and two molecules being excluded due to the AD. The most likely molecules were NC50 (*p* = 81%), NC60 (*p* = 79%), NC63 (*p* = 78%), NC67 (*p* = 77%) and NC52 (*p* = 73%). NC53, NC40, NC36, NC30, and NC31 displayed activity probabilities between 62 and 68%.

Using the Dragon7 descriptors, the model classified 37 molecules as potentially active, with five molecules having a probability in the range of 60-63% and the remaining 32 molecules having activity probability above 50%. Seven compounds did not show reliability within the applicability domain. NC52 (*p* = 63%), NC62 (*p* = 63%), NC50 (62%), NC53 (*p* = 60%), and NC39 (60%) were the compounds with the highest activity. In addition to these compounds, high probabilities were recorded for NC36 (54%), NC30 and NC40 (54%), NC51 (53%), NC31 (52%), as well as NC41 and NC34 (50%). Therefore, these compounds were also selected for synthesis and testing.

After observing the classification patterns of the models, we noticed the importance of bulky groups in the VolSurf descriptors, since the NC50 molecule (*p* = 81%) presents a bis-benzyl substitution, which refers to the influence of globularity descriptors. With the Dragon 7.0 descriptors, the influence of acid groups with carboxylic acids and amine groups is noticeable, since the compound NC52 (*p* = 63%) was the most likely in this descriptor set, and delivers values above 60% in VolSurf descriptors.

In the consensus model, a total of 23 molecules showed activity percentages above 50%. The molecules with the highest probability within the applicability domain were: NC50 (*p* = 70%), NC52 (*p* = 67%), NC67 (*p* = 66%), NC53 (*p* = 64%) and NC58 (*p* = 63%). NC40, NC36, NC30, and NC31 (56% < *p* < 60%) were also included in testing due to their easy obtainment.

### 3.3. VolSurf Descriptor Principal Component Analysis

Consensus principal component analysis (CPCA) calculates the weight of descriptor blocks and can provide insight into the most influential descriptors. The LOGS (solubility descriptors), OH2 (descriptors of hydrophilic regions), and SIZE/SHAPE (topology descriptors) were most prominent in this test set.

The PCA analysis determined that NC63, NC75, NC15, and NC27 presented structural differences compared to others in the study. The PCA corroborates the data from the CPCA, in which three groups of descriptors from the OH2 block largely explain the variability in the data through the polarity of the analyzed molecules. The most influential descriptors were identified as hydrophilic volume descriptors W1-W8, hydrophilic interaction descriptors IW1-IW4, and hydrophilic capacity descriptors CW1-CW8. Again, further analysis of these results suggests the best-performing molecules have hydrophilic regions and H-bond acceptors.

### 3.4. Toxicity Risk and Bioavailability In Silico Calculations

Toxicity is understood to be the inherent ability of a substance to produce harmful effects in a living organism or ecosystem. The toxic risk then, is the probability that the harmful effect, or toxic effect, occurs under the conditions of use of the substance [53,54].

Compounds with activity probabilities above 60% were evaluated for toxicity in the OSIRIS program. Parameters evaluated included mutagenicity, skin irritability and reproductive toxicity. The results are shown in Appendix A, and due to their essential roles in human and animal health, must be considered in the search for new therapeutic agents [55,56]. All molecules passed these standards and were therefore maintained.

In addition to evaluating the risk of toxicity, we evaluated the rate of oral absorption and violations of Lipinski’s rule, as shown in Appendix A. Absorption, both transcellular and paracellular, is an important parameter when searching for drugs and medications; absorption is influenced by the molecular properties of solubility and lipophilicity [57,58,59].

Lipinski’s rule, also known as the “rule of five,” was one of the first parameters to correlate physicochemical properties with Absorption, Distribution, Metabolism and Excretion (ADME) parameters. This parameter presents a relationship between pharmacokinetic and physicochemical properties, indicating that a molecule will have high potential as a drug if it resembles other existing drugs. The accepted parameters include a molar mass less than 500 g/mol^−1^, no more than five hydrogen bond donors or 10 hydrogen bond acceptors, and an Octanol/water partition coefficient (LogP) less than 5 [60,61].

All compounds studied had absorption rates above 60% and three compounds showed only a single violation of Lipinski’s rule, these being NC70, NC73 and NC74 that present violation referring to molar mass above 500 g/mol^−1^. NC69 and NC76 presented two violations of Lipinski’s rule, with molecular mass greater than 500 g/mol^−1^, as well as Octanol/water partition coefficient (LogP) values greater than 5, however were maintained as the pool of similar analogues displayed good absorption and bioavailability potential.

### 3.5. Selection of Molecules for Organic Synthesis

Following the in silico screening, 16 compounds were selected for organic synthesis and in vitro testing. The main selection criterion was a probability of activity above 50% in the consensus model, but the organic feasibility was also considered. The selected compounds were NC01, NC02, NC07, NC08, NC09, NC13, NC18, NC19, NC30, NC31, NC34, NC36, NC40, NC41, NC51 and NC53. The compounds were identified using the same ID as in silico and probabilities, and are shown in Appendix A.

The demonstrated probabilities correspond to the result of the consensus of the Dragon 7.0 and VolSurf descriptors. Even though some molecules did not reach 50% activity probability in consensus, this result was maintained, as in some situations the compounds reached 50% in individual models. Similarly with the applicability domain, some molecules were maintained due to being established in the spaces comprising the domain in one of the models. Some compounds were maintained to better our understanding of the characteristics of the series under study, as well as to experimentally validate the computational methodology developed.

### 3.6. In Vitro Activity Assessment

The synthesized compounds were evaluated based on their ability to inhibit the promastigote form of four types of *Leishmania*; the results are shown in the table below (Table 3). Additionally, the results of the in vitro tests are included for comparison against the predicted percentage of activity.

The performance of virtual screening studies with selenium compounds is commonly adopted in our study group, with reliable and satisfactory results, as evident in Huang’s (2021) [21] selection of selenium and chlorine substituted antileishmanial compounds. In addition to the aforementioned study, the research developed by Luís et al. (2019) [16] used a combined methodology to evaluate the antileishmanial activity of selenoglycolicamide derivatives.

The results obtained from the models were corroborated by the results of the in vitro study, in which selenium compounds NC30, NC31, NC34, NC36, NC40, NC41, NC51, and NC53 presented satisfactory inhibitory concentration minimums (IC_50_) (Table 3)**.** This suggests that these compounds may be able to exert leishmanicidal activity by inhibiting the promastigote form; however, further studies are needed to clearly elucidate the mechanism of action. Just the same, the in vitro results validate the developed model and recommend its reliability.

### 3.7. Molecular Dynamics

After analyzing the activity potential of the test compound NC34 against the four important *Leishmania* species under study, molecular dynamics simulations were performed with the test compound NC34 to assess the flexibility of the enzyme and the stability of interactions in the presence factors such as solvent, ions, pressure, and temperature. This information is important because it complements the results of docking and the biological tests, in addition to allowing the evaluation of the compounds that remained strongly bound to the studied enzymes in the presence of factors that are found in the host organism. For this, the following enzymes were chosen for analysis: CYP-51 from *Leishmania infantum* (PDB: 3L4D) (A) and Dihydroorotate Dehydrogenase from *Leishmania amazonensis* (B). NC34 showed a greater affinity for these proteins, corroborating the previous in silico and in vitro results of a greater inhibition for these two species. Then, the RMSD was calculated for the Cα atoms of the complexed enzyme and the structures of each ligand, separately.

For the CYP-51 target (PDB: 3L4D), the analysis of the RMSD metric of the protein (Figure 4A) showed that the simulation stabilization for this enzyme (red line) occurred around the period of 2 ns to the period of 7ns with RMSD values corresponding to 0.27 nm. After 7ns, and until 10 ns, the enzyme presented fluctuations in the RMSD values that corresponded to 0.30 returning to 0.27 nm. With NC34 (green line), the simulation stabilization occurred around 3ns until the end of the 4ns simulation, with RMSD values corresponding to 0.35 nm and remaining stable until the total simulation time, thus demonstrating that forms greater stability when compared to the PDB linker Fluconazole. Fluconazole presented higher RMSD values compared to NC34 and in addition, it presented high peaks after the 6 ns period, demonstrating instability. The stability of the CYP-51 protein (PDB: 3L4D) is essential to keep compounds bound to the active site.

For the Dihydroorotate dehydrogenase from *L. amazonensis* (Figure 4B), a similar behavior was observed between the three complexes. With very close RMSD values, the three compounds began to stabilize around 1 ns. The Dihydroorotate dehydrogenase complex (red) fluctuations were observed between 5 ns and 10 ns, reaching RMSD values of 0.35 nm. For NC34, fluctuations were observed in the period of 3 to 4 ns with RMSD values of 0.34 nm and in the period of 7 ns with RMSD values of 0.37 nm. After 8 ns, there was a reduction of RMSD to 0.3 nm and reduction of fluctuations. For the control ligand (blue) fluctuations occurred at the beginning of the simulation between the period of 1 to 2 ns with RMSD values of 3.0 nm, and after this period fluctuations were observed at 5 and 9 ns with RMSD values of 0.37 nm. Thus, it can be concluded that the compound NC34 showed good stability in the target under study. The stability of the Dihydroorotate dehydrogenase protein is essential to keep the compounds bound to the active site.

When analyzing the stability of the ligands in the presence of solvents in the CYP-51 enzyme (PDB: 3L4D) (Figure 5A), it was verified that the test compound NC34 presented higher RMSD values than the results obtained by the PDB ligand Fluconazole. Regarding the Dihydroorotate dehydrogenase (Figure 5B), the compound NC34 showed lower RMSD values when compared to the homologous enzyme ligand. Therefore, in the presence of solvents, ions and other factors, the compound NC34 seems to be capable of establishing strong bonds with the active site of the enzyme Dihydroorotate dehydrogenase from *L. amazonensis*. Thus, it is suggested that NC34 tends to remain in the active site even in the presence of different factors such as temperature, pressure, solvent, and ions.

To understand the flexibility of residues and amino acids that contribute to the conformational change in CYP-51 (PDB: 3L4D) (Figure 6A) and Dihydroorotate dehydrogenase (Figure 6B) enzymes, root mean square fluctuations (RMSF) were calculated for each amino acid in the protein. Residues with high RMSF values suggest more flexibility and low RMSF values reflect less flexibility. Knowing that amino acids with fluctuations above 0.3 nm contribute to the flexibility of the channel structure, we identified that among the amino acids present in the CYP-51 protein (PDB: 3L4D) (Figure 6A), the residues at positions 250, 251, 252, 253, 254 and 255 (Figure 6B) showed RMSF values above 0.3 nm, while for the Dihydroorotate dehydrogenase protein, the residues at position 310, 311, 312 and 313 contribute to the conformational change of the protein complexed to NC34. Despite this, the residues mentioned are not components of the active site of the proteins under study, which likely contributes to NC34 remaining in the active site.

## 4. Conclusions

Sixteen selene-ethylenelactamides derivatives with the highest probability of antileishmanial activity were prioritized in silico. In addition to elucidating potential binding mechanisms, multiple physicochemical properties were calculated. The screened compounds were predicted to have high oral absorption (>60%), and were largely in accord Lipinski’s rule of five, indicating good bioavailability; only seven compounds showed indications of toxicity in one or even two parameters.

Compounds NC01, NC02, NC07, NC08, NC09, NC13, NC18, NC19, NC30, NC31, NC34, NC36, NC40, NC41, NC51, and NC53 (eight *N*-arylpropanamide derivatives and eight selene-ethylenelactamide compounds), were screened in vitro and all were found to be active against at least one of the four tested forms of *Leishmania*: *Leishmania infantum, Leishmania amazonensis, Leishmania major, and Leishmania (Viannia) braziliensis*.

## 5. Patents

The main compound of this work was patented as an unprecedented substance for tuberculosis. The process was submitted to the National Institute of Industrial Property (INPI) through the INOVA/UFPB agency, having been accepted on December 22, 2021 under case number: BR 10 2021 026090 4 and petition: 870210119497.

## Figures and Tables

**Figure 1 pathogens-12-00136-f001:**
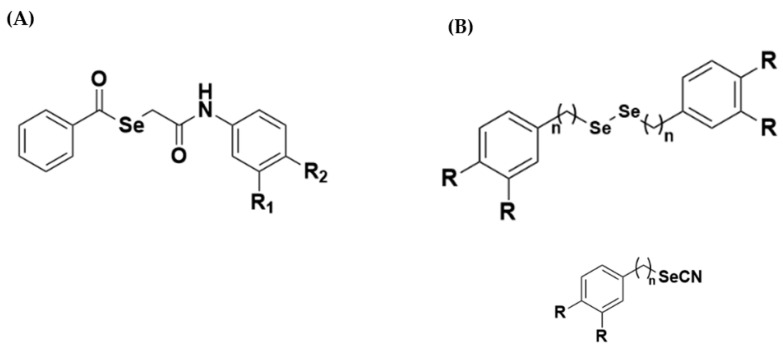
Chemical structure of selenoglycolicamide compound (**A**), as well as the selenocyanate and diselenide derivatives (**B**) that served as the inspiration for this work. Source: Adapted from Luis and Collaborators (2019) and Plano and Collaborators (2011) [16,17].

**Figure 2 pathogens-12-00136-f002:**
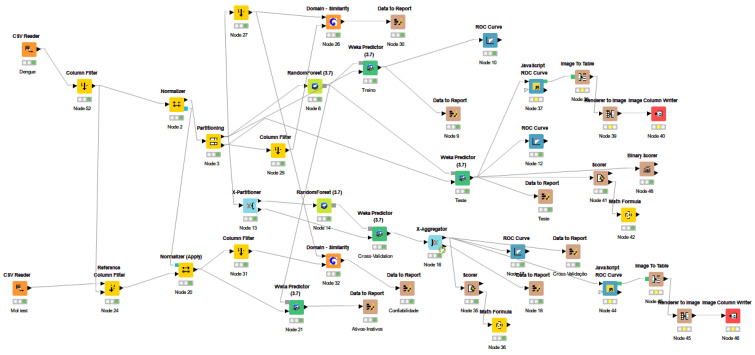
Work plan prepared in KNIME 3.2.6 software to calculate statistics.

**Figure 3 pathogens-12-00136-f003:**
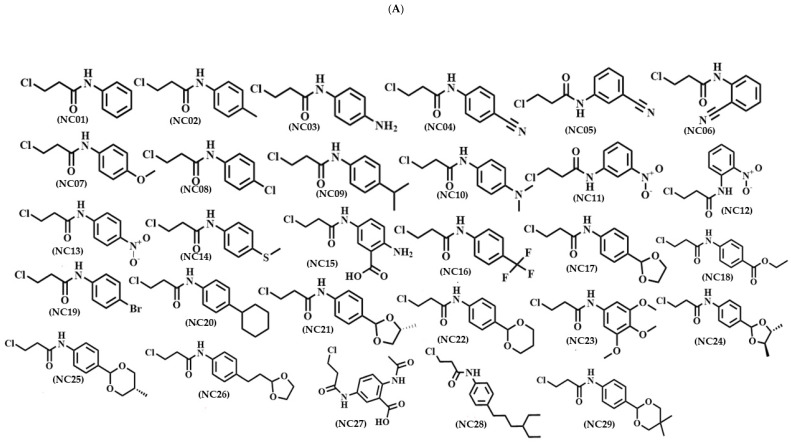
The structure of each compound. (**A**) *N*-Aryl-Propanamides derivatives and (**B**) Selene-ethylenelacticamides derivatives.

**Figure 4 pathogens-12-00136-f004:**
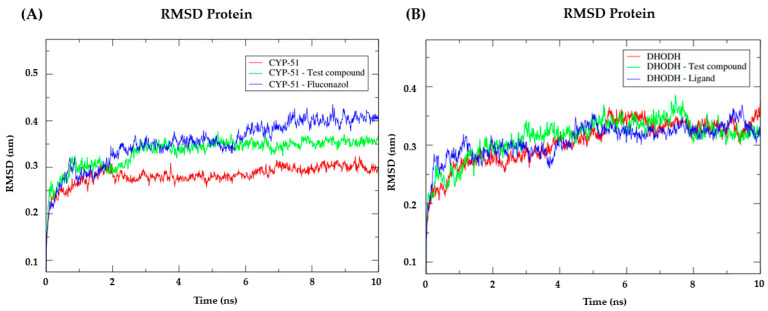
(**A**) RMSD of the Cα atoms of the enzyme CYP-51 (PDB: 3L4D) (red) and complexed to the compounds NC34 (green) and Fluconazole (blue). (**B**) RMSD of the Cα atoms of the enzyme Dihydroorotate dehydrogenase (red) and complexed to the compounds NC34 (green) and Ligand (blue).

**Figure 5 pathogens-12-00136-f005:**
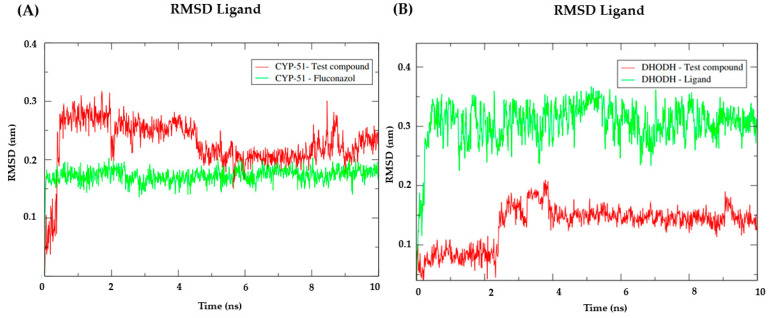
RMSD values of the Cα atoms of NC34 and control ligands. (**A**) CYP-51 (PDB: 3L4D); (**B**) Dihydro-orotate dehydrogenase.

**Figure 6 pathogens-12-00136-f006:**
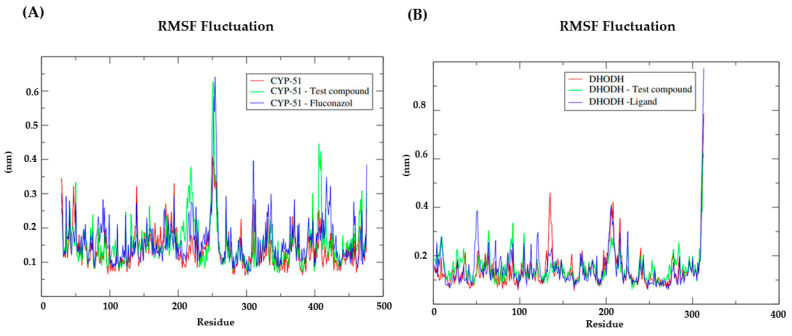
Root-mean-square fluctuation (RMSF) for the Cα atoms of the: (**A**) CYP-51 (PDB: 3L4D) enzyme complexed with the NC34 and the PDB ligand Fluconazol; (**B**) Diidroorotato desidrogenase enzyme complexed with the NC34 and the control ligand.

**Table 1 pathogens-12-00136-t001:** Summary of parameters corresponding to the results obtained for all models with Dragon 7 descriptors.

Specie	Validation	Specificity	Sensitivity	Accuracy	PPV	NPV	MCC	ROC
*L. amazonensis*	Test	0.82	0.82	0.82	0.82	0.82	0.64	0.920
	Cross	0.761	0.707	0.742	0.707	0.778	0.524	0.846
*L. infantum*	Test	0.885	0.815	0.849	0.815	0.885	0.70	0.904
	Cross	0.733	0.785	0.759	0.785	0.733	0.519	0.849
*L. braziliensis*	Test	0.79	0.85	0.821	0.85	0.79	0.642	0.878
	Cross	0.885	0.704	0.792	0.704	0.885	0.597	0.877
*L. major*	Test	0.885	0.873	0.879	0.873	0.885	0.758	0.932
	Cross	0.846	0.815	0.83	0.815	0.849	0.661	0.882

**Table 2 pathogens-12-00136-t002:** Summary of parameters corresponding to the results obtained for all models with VolSurf descriptors.

Specie	Validation	Specificity	Sensitivity	Accuracy	PPV	NPV	MCC	ROC
*L. amazonensis*	Test	0.713	0.786	0.75	0.762	0.737	0.50	0.854
	Cross	0.745	0.755	0.75	0.755	0.745	0.50	0.841
*L. infantum*	Test	0.762	0.841	0.802	0.841	0.762	0.605	0.887
	Cross	0.846	0.667	0.755	0.667	0.846	0.52	0.853
*L. braziliensis*	Test	0.786	0.844	0.816	0.825	0.806	0.632	0.897
	Cross	0.731	0.778	0.755	0.764	0.745	0.509	0.860
*L. major*	Test	0.816	0.811	0.814	0.815	0.813	0.628	0.897
	Cross	0.82	0.794	0.806	0.806	0.613	0.613	0.876

**Table 3 pathogens-12-00136-t003:** In vitro activity results of the compounds studied and comparison with the probabilities identified in the prediction model.

ID	*L.b* Pro (p%)	IC_50_	*L.i* Pro (p%)	IC_50_	*L.m* Pro (p%)	IC_50_	*L.a* Pro (p%)	IC_50_
NC01	8%	**17.6**	**59%**	**26.89**	26%	>50	18%	>50
NC02	5%	>50	**59%**	>50	32%	>50	19%	>50
NC07	5%	>50	**58%**	>50	37%	>50	10%	>50
NC08	4%	>50	**61%**	>50	32%	>50	26%	>50
NC09	16%	>50	**58%**	>50	32%	>50	19%	>50
NC13	9%	>50	**52%**	>50	42%	>50	8%	>50
NC18	25%	>50	**54%**	>50	46%	>50	14%	>50
NC19	4%	>50	**60%**	>50	16%	>50	27%	>50
NC30	44%	**7.92**	**69%**	**10.4**	**58%**	**13.97**	**64%**	**14.5**
NC31	44%	**5.23**	**68%**	**7.36**	**56%**	**5.7**	**61%**	**15.3**
NC34	45%	**2.1**	**67%**	**3.28**	49%	**4.84**	**72%**	**7.28**
NC36	48%	**4.24**	**68%**	**8.52**	**59%**	**10.36**	**70%**	**15.9**
NC40	**58%**	**5.85**	**74%**	**6.6**	**60%**	>50	**68%**	**8.2**
NC41	**53%**	>50	**59%**	>50	46%	>50	**61%**	>50
NC51	**61%**	**3.71**	**68%**	**6.43**	47%	**9.85**	**63%**	**16.4**
NC53	**54%**	**3.9**	**71%**	**8.05**	**64%**	**8.63**	**70%**	**25.16**

**Legend:** In bold and red, the compounds with probability equal to or greater than 50% (in the columns referring to probability values) are highlighted, as well as the compounds with the best IC_50_ values.

## Data Availability

Not applicable.

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
