# Peer review of "Selene-Ethylenelacticamides and N-Aryl-Propanamides as Broad-Spectrum Leishmanicidal Agents"

_pathogens, 2023, doi:10.3390/pathogens12010136_

Round 1
Reviewer 1 Report
Dear authors,
please find enclosed my observation. There are mistakes that require particular attention.
-Please all chemical structures should be presented and well drawn.
-Systematic names or genus should be written in italics
-The chemical shifts requires more details for example if you have a doublet you may inform its coupling constant.

Author Response
|
1 |
Please all chemical structures should be presented and well drawn. |
|
|
The structures were corrected and standardized using the ChemDraw software. |
|
2 |
Systematic names or genus should be written in italics. |
|
|
Scientific names were standardized to italics. |
|
3 |
The chemical shifts requires more details for example if you have a doublet you may inform its coupling constant. |
|
|
The detailing of the chemical shifts was carried out with the insertion of the coupling constant. The description of the compounds can be found in the methodology section. The description of the synthesis of the compounds with the respective spectra was moved to the supplementary material section. |
Reviewer 2 Report
1. Although the work initially appears to be interesting, there are numerous grammatical errors in the abstract itself, including certain sentences. Leishmania infantum, Leishmania amazonensis, Leishmania major, and Leishmania (Viannia) braziliensis were used as strong in silico models to predict the biological activities of 77 selenoesters and their N-aryl-propanamide derivatives. T, needs to have its grammar fixed.
2. Authors discussed TOXICITY , but did they go into detail about the methodology used and the evaluation method used?
3. Describe the The Knime 3.6.2 software's QSAR execution process in some detail.
4. Was it necessary to conduct homology modelling? The process did not contain the enzyme's name.
5. Without the name and specifics of the enzyme and without any explanation as to why the enzyme was chosen, docking and homology modelling appear to be blind studies.
6. why author havent went for H1 NMR studies for synthesized compounds?
7. mechanistic interpretation is missing in QSAR discussion, the seems to be statistical anaysis only.
8. author fails to find the difference in the activity on the basis of molecular descriptor.
9. too many Ramachandran plots were added, i don’t know why?
10. author haven’t explained the bioactivity difference in molecular docking analysis.
11. i must suggest to add MD simulation of at least two most active molecules.
12. with the inclusion these comment, the manuscript can be recommended for publication.
Author Response
|
1 |
Although the work initially appears to be interesting, there are numerous grammatical errors in the abstract itself, including certain sentences. Leishmania infantum, Leishmania amazonensis, Leishmania major, and Leishmania (Viannia) braziliensis were used as strong in silico models to predict the biological activities of 77 selenoesters and their N-aryl-propanamide derivatives. T, needs to have its grammar fixed |
|
|
The scientific name of the species was corrected for italics, as well as the correction of the name of the compounds. |
|
2 |
Authors discussed TOXICITY , but did they go into detail about the methodology used and the evaluation method used? |
|
|
Toxicity analysis was done via in silico toxicity prediction and evaluation of ADMET parameters by OSIRIS 5.0 software. The summary, methodology, and results were changed to reflect this. |
|
3 |
Describe the The Knime 3.6.2 software's QSAR execution process in some detail. |
|
|
The characteristics of the databases and modeling were described. In addition, a figure demonstrating the procedure was added. |
|
4 |
Was it necessary to conduct homology modelling? The process did not contain the enzyme's name. |
|
|
The mechanisms are necessary because they refer to the survival of the parasite. The description of the PDS's that served as a framework for the construction was carried out, however the resulting data were moved to supplementary material. |
|
5 |
Without the name and specifics of the enzyme and without any explanation as to why the enzyme was chosen, docking and homology modelling appear to be blind studies. |
|
|
The justification and a brief description of the mechanisms of each chosen enzyme was added. Data have been added to the Results section under Molecular Docking, but has been removed from supplemental material. |
|
6 |
why author havent went for H1 NMR studies for synthesized compounds? |
|
|
The chemical shifts of all compounds are described in the Methodology section. The description and explanation of the H1 NMR analyses with all spectra are found in the section on Organic Synthesis that was removed for supplementary material. |
|
7 |
mechanistic interpretation is missing in QSAR discussion, the seems to be statistical anaysis only. |
|
|
Information describing existing differences in the classification performed by the descriptors was added. |
|
8 |
author fails to find the difference in the activity on the basis of molecular descriptor. |
|
|
Information describing existing differences in the classification performed by the descriptors was added. |
|
9 |
Lots of Ramachandran graphics added, I don't know why? |
|
|
Ramachandram charts were added as 12 engines were built so they were used for validation. As enzymes are essential for parasite survival, we chose to build them all. The description of the rationale for the choice of enzymes, as well as the entire discussion regarding homology, has been removed from the supplementary material. |
|
10 |
author haven’t explained the bioactivity difference in molecular docking analysis. |
|
|
A description of the ligand-protein interactions was added, as well as a comparison with the interactions described in the reference article of the Protein Data Bank library. In addition, we added the description of the groups that contribute to the interaction with the enzymes, however the molecular Docking results were moved to the Supplementary Material section. |
|
11 |
i must suggest to add MD simulation of at least two most active molecules. |
|
|
Molecular dynamics simulation was carried out with the compound NC34, as presenting the best results in the experimental validation. The evaluated mechanisms corresponded to the enzyme CYP-51 (PDB: 3L4D) from Leishmania infantum and the enzyme Dihydroorotate dehydrogenase from Leismania amazonensis, as the compound NC34 showed affinity values close to those demonstrated by the reference ligand. |
|
12 |
with the inclusion these comment, the manuscript can be recommended for publication. |
|
|
All the comments were carefully addressed in the text. We would like to appreciate the Reviewer for such valuable comments that helped us to greatly improve the quality of the manuscript. |
Reviewer 3 Report
The authors present the synthesis and in vitro characterization of a series of selene-ethylenelacticamides and N-Aril-Propanamides as broad-spectrum leishmanicidal agents, selected by QSAR modeling. The manuscript will be of interest to those working in computer aided drug design, as well to those involved in the development of leishmanicidal agents. However, there are some points to be considered.
1.- In general, the manuscript is well done and interesting, but in the case of the homology modeling and docking sections, is not clear the utility of these data, at least in the context of the work. I agree with the aim to know the potential targets of the compounds, but here, the targets were selected previously and just the data serve to know in which of these targets the compounds bound better but there could be another protein or enzyme important in leishmania where compounds bind with a higher score. In conclusion, these part of the manuscript does not help to increase the quality of the manuscript, to the contrary, only disperse or distract from the principal idea of the work, therefore, it would be desirable to eliminate these sections. The rest of the work is enough and sufficiently scientific sound to deserve its publication.
2.- Sections 3.9 and 3.10 (including 3.10.1 and 3.10.2) should be modified to maintain just the most important in reference to the synthesis and characterization, a lot of the information described can be resumed in methodology and is included in supplementary material.
3.- In the abstract, is important to state that the ADMET properties were predicted because in its actual form seems like they were calculated experimentally.
4. A minor detail, the language at the end of the legend in Table 05 is not English.
Author Response
|
1 |
In general, the manuscript is well done and interesting, but in the case of the homology modeling and docking sections, is not clear the utility of these data, at least in the context of the work. I agree with the aim to know the potential targets of the compounds, but here, the targets were selected previously and just the data serve to know in which of these targets the compounds bound better but there could be another protein or enzyme important in leishmania where compounds bind with a higher score. In conclusion, these part of the manuscript does not help to increase the quality of the manuscript, to the contrary, only disperse or distract from the principal idea of the work, therefore, it would be desirable to eliminate these sections. The rest of the work is enough and sufficiently scientific sound to deserve its publication. |
|
|
Molecular docking and homology data were removed for supplemental material section. |
|
2 |
Sections 3.9 and 3.10 (including 3.10.1 and 3.10.2) should be modified to maintain just the most important in reference to the synthesis and characterization, a lot of the information described can be resumed in methodology and is included in supplementary material. |
|
|
This section has been summarized and removed for supplemental material. Only the data referring to the compounds in the Methodology section remained. |
|
3 |
In the abstract, is important to state that the ADMET properties were predicted because in its actual form seems like they were calculated experimentally. |
|
|
The description was changed to reflect the in silico methodology |
|
4 |
A minor detail, the language at the end of the legend in Table 05 is not English. |
|
|
The subtitle has been corrected. |
Reviewer 4 Report
Manuscript ID-pathogens-2046240 is an extensive work aimed at evaluating the antileishmanial activity of a series of 48 new selenoethylenelacticamides and 29 already known N-arylpropanamides by combining several in silico approaches (QSAR, homology modeling, molecular docking, ADMET, etc.) and in vitro antiparasitic essays. Although this is a systematic study with promising results for the antiparasitic activity of some ligands tested, the article has several deficiencies that need to be fully addressed.
Line 23: Specify if the half a million cases are new or accumulated
Line 31: Change "Further results" to "ADMET results"
Line 32: Define if the toxicity evaluation was theoretical or experimental
Line 34: How was the biological activity confirmed and what were the best results and why?
Line 41: These or This?
Lines 48 and 50: L. infantum should be italicized, same for all other strains. Unify this throughout the document.
Line 57: Better defines in which organ the promastigotes exert rapid migration and dissemination and how.
Line 62: Add the appropriate citations after the word "endemics"
Lines 70–71: Why is mitefosine unavailable in Brazil? please better support this statement.
Lines 72–83: The structures that served as inspiration for this work must be presented in a Figure.
Line 98: Add the appropriate reference citation after the word "literature"
Line 104–105: how was the conformational minimization dealt with? that is, how was the lowest energy isomer guaranteed?
Line 124: here fourteen homology models are mentioned, but in the results twelve homologies are made, this discrepancy must be corrected.
Line 150: The formula used in Molegro for calculating the scoring function must be provided in the document as an equation.
Lines 163: Subscripts in chemical formulas will need to be reviewed and corrected throughout the document.
Lines 165, 171, 375, 558, 589: Extra space must be removed.
Line 176: Change "1x106" to "1x106"
Lines 189, 195, 201, 208, 214, 221, 227, 234, 242, 250, 258, 267, 276, 285, 293, 302: Change from "Alifático" to "aliphatic"
Lines 259, 277: dmso or DMSO? unify throughout the document.
Line 240: This section should start with a paragraph clarifying why these 16 molecules were synthesized and not others.
Line 312: In this section, a Figure containing all 77 derivative structures must be included. Consequently, Figures 1 to 4 should be removed.
Line 331: "both descriptors" is an ambiguous phrase, please define better its meaning.
Lines 504, 506, 509, 512: Ramachandran graphs should be sent to the supporting information.
Line 524: Change "2.4" to "3.4"
Lines 549–551: Delete this paragraph
Line 741: Please further explain why most of the compounds violated one Lipinski's rule and why NC74 violated two.
Line 744–746: This section should be moved close to the QSAR section.
Line 762: Table 04 must be eliminated since its data are also reported in Table 05.
Line 774–848. all information provided in sections 3.9 and 3.10 should be summarized and sent to the materials and methods section.
Line 849–926: All information provided by sections 3.10.1 and 3.10.2 is superfluous and should be sent to support information.
Author Response
|
1 |
Line 23: Specify if the half a million cases are new or accumulated |
|
|
The specification has been made. |
|
2 |
Line 31: Change "Further results" to "ADMET results" |
|
|
Changed |
|
3 |
Line 32: Define if the toxicity evaluation was theoretical or experimental |
|
|
It was experimental using OSIRIS 5.0 software. Changed. |
|
4 |
Line 34: How was the biological activity confirmed and what were the best results and why? |
|
|
The procedure was described and explained. |
|
5 |
Line 41: These or This? |
|
|
Changed. |
|
6 |
Lines 48 and 50: L. infantum should be italicized, same for all other strains. Unify this throughout the document. |
|
|
Changes have been made to the entire article. |
|
7 |
Line 57: Better defines in which organ the promastigotes exert rapid migration and dissemination and how. |
|
|
The description was added. |
|
8 |
Line 62: Add the appropriate citations after the word "endemics" |
|
|
The citation has been inserted |
|
9 |
Lines 70–71: Why is mitefosine unavailable in Brazil? please better support this statement. |
|
|
Mitelfosine is used within the Unified Health System (SUS), its insertion was defined in 2018. This information has been corrected. |
|
10 |
Lines 72–83: The structures that served as inspiration for this work must be presented in a Figure. |
|
|
Structures have been added. |
|
11 |
Line 98: Add the appropriate reference citation after the word "literature" |
|
|
Quotes have been added. |
|
12 |
Line 104–105: how was the conformational minimization dealt with? that is, how was the lowest energy isomer guaranteed? |
|
|
The structures were standardized in the ChemAxon Standardizer program. A justification for using this program was added to the manuscript, as well as its function. |
|
13 |
Line 124: here fourteen homology models are mentioned, but in the results twelve homologies are made, this discrepancy must be corrected. |
|
|
The correction was made. |
|
14 |
Line 150: The formula used in Molegro for calculating the scoring function must be provided in the document as an equation. |
|
|
The formula has been added. |
|
15 |
Lines 163: Subscripts in chemical formulas will need to be reviewed and corrected throughout the document. |
|
|
Chemical structures and molecular formulas have been standardized and corrected. |
|
16 |
Lines 165, 171, 375, 558, 589: Extra space must be removed. |
|
|
The space has been removed. |
|
17 |
Line 176: Change "1x106" to "1x106" |
|
|
Changed. |
|
18 |
Lines 189, 195, 201, 208, 214, 221, 227, 234, 242, 250, 258, 267, 276, 285, 293, 302: Change from "Alifático" to "aliphatic" |
|
|
Changed. |
|
19 |
Lines 259, 277: dmso or DMSO? unify throughout the document. |
|
|
Changed. |
|
20 |
Line 240: This section should start with a paragraph clarifying why these 16 molecules were synthesized and not others. |
|
|
A paragraph was added with the justification of the molecules that were synthesized. |
|
21 |
Line 312: In this section, a Figure containing all 77 derivative structures must be included. Consequently, Figures 1 to 4 should be removed. |
|
|
The figure with the 77 compounds has been added, and figures 01-04 have been moved to supplemental material. |
|
22 |
Line 331: "both descriptors" is an ambiguous phrase, please define better its meaning. |
|
|
Changed. |
|
23 |
Lines 504, 506, 509, 512: Ramachandran graphs should be sent to the supporting information. |
|
|
This section has been moved to supplemental material. |
|
24 |
Line 524: Change "2.4" to "3.4" |
|
|
Changed. |
|
25 |
Lines 549–551: Delete this paragraph |
|
|
Deleted the paragraph. |
|
26 |
Line 741: Please further explain why most of the compounds violated one Lipinski's rule and why NC74 violated two. |
|
|
The explanation was performed. |
|
27 |
Line 744–746: This section should be moved close to the QSAR section. |
|
|
The order has been changed. |
|
28 |
Line 762: Table 04 must be eliminated since its data are also reported in Table 05. |
|
|
Table 04 has been moved to Supplementary Material. |
|
29 |
Line 774–848. all information provided in sections 3.9 and 3.10 should be summarized and sent to the materials and methods section. |
|
|
The information has been summarized and moved to supplemental material. |
|
30 |
Line 849–926: All information provided by sections 3.10.1 and 3.10.2 is superfluous and should be sent to support information. |
|
|
The information has been summarized and moved to supplemental material. |
Round 2
Reviewer 2 Report
Accept in present form
Reviewer 3 Report
The manuscript was corrected according to the suggestions. Therefore, I recommend its publication.
Reviewer 4 Report
The authors have significantly improved the quality of the manuscript and have adequately addressed all my comments. Now the document presents sufficient scientific quality to be accepted for publication in the special issue "Contribution of Computational Tools for Drug Development against Pathogens" in the Pathogens journal.